# TAMMs: Change Understanding and Forecasting in Satellite Image Time Series with Temporal-Aware Multimodal Models

**Zhongbin Guo[1], Yuhao Wang[1], Ping Jian[1],\*Chengzhi Li[1], Xinyue Chen[1], Zhen Yang[1], Ertai E[2]**
[1]School of Computer Science & Technology, Beijing Institute of Technology
[2]School of Computing, National University of Singapore
guozhongbin@bit.edu.cn pjian@bit.edu.cn

## Abstract

Temporal Change Description (TCD) and Future Satellite Image Forecasting (FSIF) are critical, yet historically disjointed tasks in Satellite Image Time Series (SITS) analysis. Both are fundamentally limited by the common challenge of modeling long-range temporal dynamics. To explore how to improve the performance of methods on both tasks simultaneously by enhancing long-range temporal understanding capabilities, we introduce **TAMMs**, the first unified framework designed to jointly perform TCD and FSIF within a single MLLM-diffusion architecture. TAMMs introduces two key innovations: Temporal Adaptation Modules (**TAM**) enhance frozen MLLM's ability to comprehend long-range dynamics, and Semantic-Fused Control Injection (**SFCI**) mechanism translates this change understanding into fine-grained generative control. This synergistic design makes the understanding from the TCD task to directly inform and improve the consistency of the FSIF task. Extensive experiments demonstrate TAMMs significantly outperforms state-of-the-art specialist baselines on both tasks.

## 1 Introduction

Understanding temporal change and forecasting future scenes from Satellite Image Time Series (SITS) is fundamental to critical applications Fu et al. (2024); Lara-Alvarez et al. (2024), yet it poses a profound challenge: predicting the future requires a deep understanding of the past change. While foundation models like Multimodal Large Language Models (MLLMs) Wu et al. (2024); Bai et al. (2025) and diffusion-based generators Sastry et al. (2024); Liu et al. (2025a) have demonstrated remarkable capabilities, their application to this domain remains fragmented and fundamentally limited. As illustrated in Fig. 1, the field is split into disjointed paradigms: static image captioning that ignores temporal context Bazi et al. (2024), and temporal description methods that lack semantic depth of MLLMs Peng et al. (2024); similarly, generative approaches are divided between single-image synthesis that cannot model evolution Liu et al. (2025a), and temporal forecasting struggles with long-term consistency because they fail to grasp semantic drivers of change Khanna et al. (2024). This division reflects a scientific gap —— the absence of a **unified framework** of temporal-aware multimodal model for satellite image change understanding and forecasting. To address this, We present TAMMs which can reason about **why** changes occur and use that reasoning to generate **what** will happen next.

*How can model reasons about why these temporal changes occur?* Existing Temporal Change Description methods fuse temporal multi-graph information through simple interactions, and their long-range temporal reasoning capabilities are limited Peng et al. (2024). To bridge this gap, we introduce novel, parameter-efficient approach to awaken the latent temporal reasoning abilities of frozen MLLM through **Temporal Adaptation Modules (TAM)**, designed to both inject explicit awareness of arbitrary time intervals and guide the model's focus onto salient change narratives, enhancing its performance in long-range temporal change reasoning.

*How can model use temporal change reasoning to generate what will happen next?* As existing methods use only text guidance and lack fine-grained temporal semantic interaction Khanna et al.

---

*Corresponding Author.

(2024), we introduce a novel mechanism to bridge this understanding with generation: **Semantic-Fused Control Injection (SFCI)**, which converts MLLM's abstract, temporally-grounded semantic features into precise, multi-scale control signals. It adaptively fuses these semantic signals with low-level structural priors using trainable gated fusion units, directly guiding the diffusion model's denoising process, which ensures forecasted images are not just plausible standalone pictures, but temporally consistent continuations of historical trajectory observed.

This synergistic coupling of temporal-aware MLLM (via TAM) and consistency-aware diffusion model (via SFCI) forms our unified framework, which we name **TAMMs**. The framework is designed to produce two synergistic outputs: high-quality temporal change description from the MLLM, and temporally consistent future image forecast from the guided diffusion model.

While both tasks can be assessed with standard metrics, rigorously evaluating the *temporal consistency* of forecast is a significant challenge, as existing metrics suffer from an evaluation gap in this regard. To address this, we introduce the **Temporal Consistency Score (TCS)**, the first metric designed to specifically quantify the consistency of predicted changes against historical dynamics. Extensive experiments demonstrate TAMMs significantly outperforms all baselines on both tasks, with its superiority in forecasting particularly highlighted by our TCS metric.

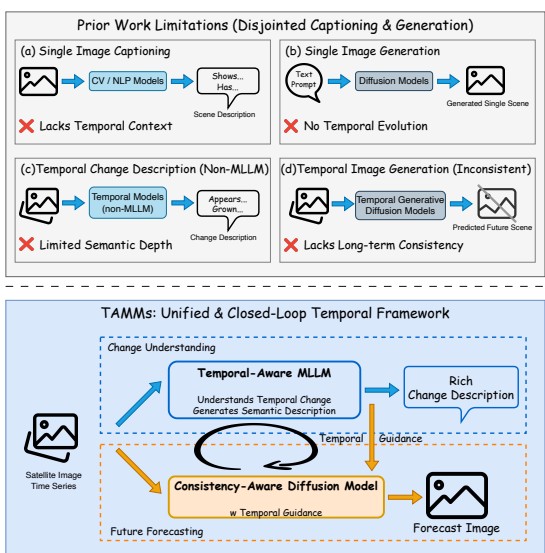

Figure 1: Our TAMMs framework unifies temporal change understanding and future forecasting, using a temporal-aware MLLM to guide a diffusion model in a synergistic process.

**Our main contributions are summarized as follows:**

- We propose **TAMMs**, the first unified framework to synergistically couple temporal change understanding with future forecasting in a mutually reinforcing manner.

- We introduce **TAM** to awaken latent long-range temporal reasoning abilities of MLLM for SITS.

- We design a novel **SFCI** mechanism to translate the MLLM's high-level temporal understanding into fine-grained generative control.

- We design the **TCS**, the first metric designed to specifically quantify the consistency of predicted changes against historical dynamics.

## 2 TASK DEFINITION

Our work introduces a unified framework to address two coupled, yet historically disjointed, tasks central to SITS analysis. We argue that solving them synergistically is key to unlocking more reliable, reasoning-based forecasting.

### 2.1 TASK 1: TEMPORAL CHANGE UNDERSTANDING

Given a sequence of temporally ordered satellite images, $\mathcal{I} = \{\mathbf{I}_1, \mathbf{I}_2, \ldots, \mathbf{I}_t\}$, the goal is to generate a natural language description, $D$, that accurately summarizes the key dynamic changes that have occurred over the observation period. This requires comprehending the temporal evolution, such as urban expansion or agricultural cycles. While prior work like SITSCC Peng et al. (2024) has explored this, it often lacks the deep semantic reasoning of modern MLLMs, struggling to capture the underlying causal narrative of the observed changes.

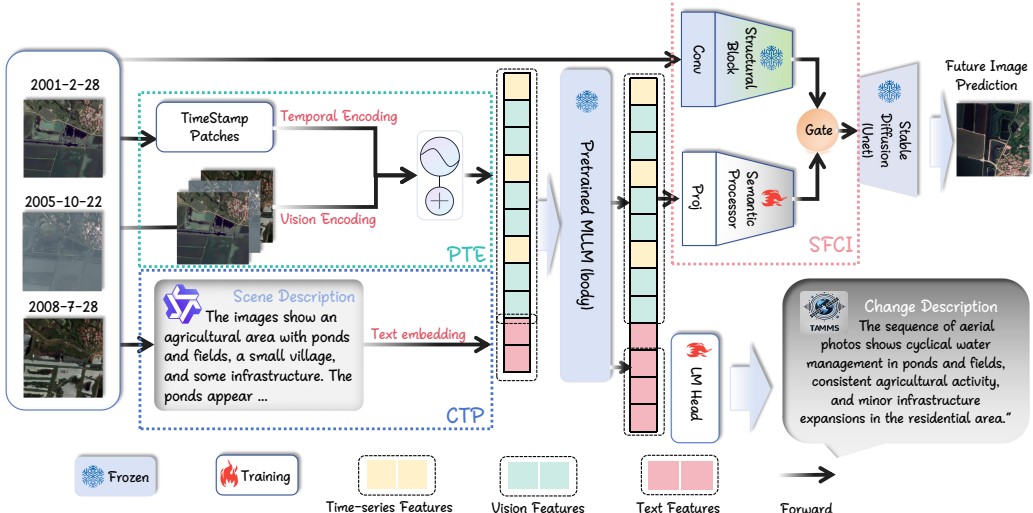

Figure 2: The TAMMs framework architecture. The temporal change understanding stage uses **Temporal Adaptation Modules (TAM)** to awaken the temporal reasoning of a frozen MLLM. The future forecasting stage then guides a frozen diffusion U-Net using our core **Semantic-Fused Control Injection (SFCI)** mechanism, which translates the MLLM's deep temporal understanding ($\mathbf{M}_t$) into multi-scale control signals. Critically, only the lightweight adapter components are trained.

## 2.2 TASK 2: TEMPORALLY CONSISTENT FUTURE FORECASTING

Given the same historical sequence $\mathcal{I}$, the goal is to generate a plausible future image $\hat{\mathbf{I}}_{t+1}$ for a subsequent time point. A critical and often overlooked requirement is that the forecast must be temporally consistent: the generated changes should represent a logical continuation of the trends and dynamics observed in the historical data. Approaches like DiffusionSat Khanna et al. (2024) primarily conditioned on metadata can only exhibit some differences in seasonal evolution lack mechanisms to ensure that the generated future is explicitly consistent with the nuanced, patch-level evolution observed in the past.

## 2.3 EVALUATION METRIC FOR TEMPORAL CONSISTENCY

A significant challenge in temporal forecasting is that standard image quality metrics (e.g., PSNR Zhang et al. (2020), SSIM Wang et al. (2004)) are insufficient, as they possess an evaluation gap: they cannot penalize temporally implausible futures. A prediction could be perceptually realistic but completely inconsistent with historical trends. To address this, we introduce a new metric.

**Temporal Consistency Score (TCS).** This metric quantifies the temporal plausibility of a forecast by comparing the spatio-temporal dynamics of predicted changes against those of the recent past. Let $\mathcal{C}(\mathbf{I}_a, \mathbf{I}_b)$ be a change detection function producing a binary change mask. We define the historical change mask as $\mathbf{M}_{hist} = \mathcal{C}(\mathbf{I}_{t-1}, \mathbf{I}_t)$ and the predicted change mask as $\mathbf{M}_{pred} = \mathcal{C}(\mathbf{I}_t, \hat{\mathbf{I}}_{t+1})$, using the P2V-CD framework Lin et al. (2023). The TCS is then formulated as the product of two scores assessing geometric consistency.

*Spatial Proximity Score (SPS)* quantifies the locational concordance between the change centroids:

$$\text{SPS}(\mathbf{M}_{hist}, \mathbf{M}_{pred}) = \exp\left(-\frac{\|\mu(\mathbf{M}_{pred}) - \mu(\mathbf{M}_{hist})\|_2}{\sigma}\right) \tag{1}$$

where $\mu(\cdot)$ is the centroid operator and $\sigma = 0.2$ is a scaling parameter.

*Area Consistency Score (ACS)* evaluates the congruence in the magnitude of change:

$$\text{ACS}(\mathbf{M}_{hist}, \mathbf{M}_{pred}) = \exp\left(-\frac{\beta \cdot ||\mathbf{M}_{pred}| - |\mathbf{M}_{hist}||}{\max(|\mathbf{M}_{pred}|, |\mathbf{M}_{hist}|) + \epsilon}\right) \tag{2}$$

where $|\cdot|$ is the area (pixel count) operator and $\beta = 1.0$.

The final composite score is the product of these two components:

$$\text{TCS} = \text{SPS} \cdot \text{ACS} \tag{3}$$

By anchoring the evaluation in the spatial distribution and magnitude of detected transformations, TCS provides a robust and discriminative measure of temporal consistency.

## 3 METHODOLOGY

As outlined in the introduction, our TAMMs framework is designed to first empower an MLLM to understand temporal dynamics, and then use this understanding to guide a generative model. This section details the core technical innovations for each part: the Temporal Adaptation Modules (TAM) for the MLLM, and the Semantic-Fused Control Injection (SFCI) for the diffusion model, which's framework can be overviewed in Figure 2.

### 3.1 AWAKENING TEMPORAL CHANGE UNDERSTANDING IN MLLMS

A key challenge is that pretrained MLLMs lack an inherent understanding of the sparse, long-term, and irregular time intervals found in SITS. However, state-of-the-art video-centric MLLMs Bai et al. (2025); Liu et al. (2025b) are typically optimized for densely sampled video data with fixed, short-term temporal intervals. Satellite Image Time Series, in contrast, where the intervals between consecutive images can span several years. This fundamental mismatch in temporal structure renders existing models incapable of accurately perceiving the gradual, long-term evolution inherent in SITS. To address this without costly full-model finetuning, we introduce Temporal Adaptation Modules (TAM), which has two complementary, parameter-efficient modules designed to "reprogram" a frozen MLLM for temporal awareness, as illustrated in Fig. 3.

**Physical Time Encoder (PTE).** To make the passage of time explicit to the MLLM, PTE injects a learnable temporal token **[TIME_DIFF]** into the input sequence. This token's embedding $\mathbf{t}_i$ is dynamically conditioned on the specific time interval $\Delta t_i$ via a small MLP, allowing it to represent arbitrary time gaps:

$$\mathbf{t}_i = \mathbf{e}_{[\text{TIME\_DIFF}]} + \text{MLP}_{\text{time}}(\phi(\Delta t_i)) \tag{4}$$

These temporal tokens are then inserted between the visual features of consecutive images $(\mathbf{V}_i, \mathbf{V}_{i+1})$, allowing the MLLM's attention mechanism to directly associate visual changes with the corresponding time interval.

**Contextual Temporal Prompting (CTP).** While PTE provides the raw temporal signal, CTP provides high-level semantic guidance. CTP provides detailed scene description, as well as using a structured textual prompt that instructs the MLLM to perform a specific temporal reasoning task: describing the changes observed in the visual sequence. This focuses MLLM's powerful but general-purpose reasoning abilities on the exact task of identifying and articulating temporal dynamics. Concrete prompt structure can be found in Appendix A.4.

**Synergistic Output.** Together, PTE and CTP enable the MLLM to produce two outputs: a textual description and, more critically, a semantic feature vector $\mathbf{M}_t$. This vector encapsulates the MLLM's temporal understanding and serves as the primary conditioning signal for the generative module. A composite loss, $\mathcal{L} = \lambda_{\text{text}}\mathcal{L}_{\text{text}} + \lambda_{\text{temp}}\mathcal{L}_{\text{temp}}$, guides the training of the adapters by balancing text accuracy with temporal regularization.

### 3.2 SEMANTIC-FUSED CONTROL INJECTION (SFCI) FOR TEMPORAL FORECASTING

Having awakened the MLLM's temporal reasoning to produce a rich semantic feature vector $\mathbf{M}_t$, the central challenge becomes translating this abstract change understanding into fine-grained control for a diffusion-based generator. Previous approach like standard ControlNet Zhang et al. (2023) use this understanding to generate a textual description to control the procedure of generation. However, this is fundamentally insufficient for high-consistency forecasting. A global text prompt, even a detailed one, is too coarse a signal to provide the necessary patch-level semantic guidance for how specific regions should evolve into the next time step, which cannot ensure the high-consistency generation of fine-grained image details required by temporal forecasting.

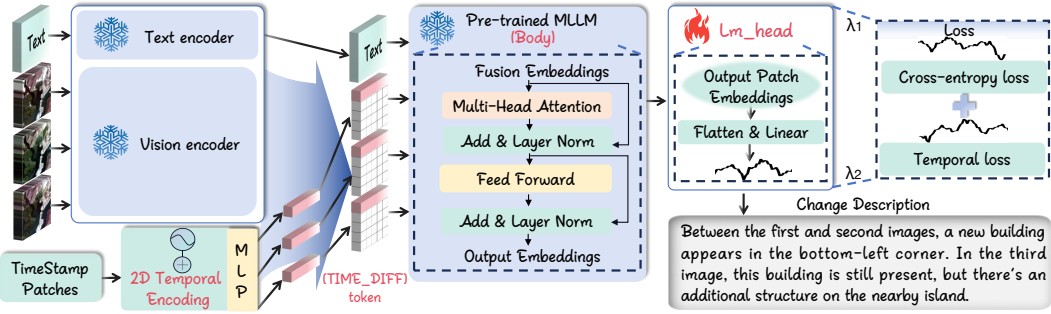

Figure 3: Temporal enhancement modules for the MLLM. **PTE** processes timestamps into learnable tokens to inject explicit temporal awareness into the visual feature stream. **CTP** (via prompting) provides high-level task guidance. Together, they enable the frozen MLLM to comprehend change narratives, which are then optimized via a composite loss.

To overcome this limitation, we introduce Semantic-Fused Control Injection (SFCI) mechanism, which is designed to bypass the bottleneck of coarse textual control. Implemented via an Enhanced Control Module (ECM) that works in parallel with a frozen diffusion U-Net Rombach et al. (2022), SFCI directly translates the MLLM's rich, multi-image temporal change understanding encapsulated in $\mathbf{M}_t$ into multi-scale, spatially-aware control signals. As shown in Fig. 4, it synergistically fuses this high-level semantic guidance with low-level structural priors, enabling precise, patch-level control that ensures the generated future is a temporally consistent evolution of the past.

**The SFCI Mechanism.** For each feature map $\mathbf{h}_l^{(enc)}$ from the $l$-th level of the U-Net encoder, SFCI extracts and adaptively fuses structural and semantic information.

(1) *Structural Path:* To capture the low-level spatio-temporal evolution, the encoder features are first processed by a frozen 3D Control Block Khanna et al. (2024), yielding a structural control signal $\mathbf{h}_l^{(ctrl)}$ that encodes underlying visual dynamics:

$$\mathbf{h}_l^{(ctrl)} = \text{CtrlBlock}_{3D}^{(l)}(\mathbf{h}_{l,ST}^{(enc)}) \in \mathbb{R}^{B \times C_l' \times T \times H_l \times W_l} \tag{5}$$

(2) *Semantic Path:* The MLLM's semantic feature $M_t$ is projected into a non-spatial intermediate representation $\mathbf{s}_l^{(proj)}$ by a level-specific processor $\text{SemProc}^{(l)}$. This is then tiled to create the spatially-aware guidance signal $\mathbf{s}_l$:

$$\mathbf{s}_l^{(proj)} = \text{SemProc}^{(l)}(\mathbf{M}_t), \qquad \mathbf{s}_l = \mathcal{T}H_l, W_l(\mathbf{s}_l^{(proj)}) \tag{6}$$

(3) *Adaptive Gated Fusion:* A dynamic gate $\mathbf{g}_l$, computed from the intermediate semantic features $\mathbf{s}_l^{(proj)}$, adaptively interpolates between the structural signal $\mathbf{h}_l^{(ctrl)}$ and the semantic guidance $\mathbf{s}_l$ to produce the fused features $\mathbf{f}_l$ :

$$\mathbf{g}_l = \mathcal{T}\{H_l, W_l\}\left(\sigma\left(\text{Gate}^{(l)}(\mathbf{s}_l^{(proj)})\right)\right) \qquad \mathbf{f}_l = (1 - \mathbf{g}_l) \odot \mathbf{h}_l^{(ctrl)} + \mathbf{g}_l \odot \mathbf{s}_l \tag{7}$$

where $\mathcal{T}\{H_l, W_l\}$ denotes a tiling operation that replicates its input tensor across the spatial dimensions $(H_l, W_l)$ to match the feature map size.

(4) *Temporal Refinement:* To model long-range dependencies, the fused features $\mathbf{f}_l$ are refined by Temporal Transformer $\text{TempTrans}^{(l)}$. As operator $\Psi_l$ first reshapes the tensor to treat the time dimension $T$ as the sequence length for the transformer. The output is then integrated via a weighted residual connection:

$$\mathbf{z}_l = \alpha_l \cdot \Psi_l^{-1}\left(\text{TempTrans}^{(l)}\left(\Psi_l(\mathbf{f}_l)\right)\right) + (1 - \alpha_l) \cdot \mathbf{f}_l \tag{8}$$

where $\alpha_l \in [0, 1]$ is a learned mixing parameter and $\Psi_l^{-1}$ is its inverse operation.

**Integration with U-Net Decoder.** The final, refined control signal $\mathbf{z}_l$ from the ECM at each level is aggregated along the time dimension to produce $\mathbf{z}_l'$. This unified, multi-scale control signal is then

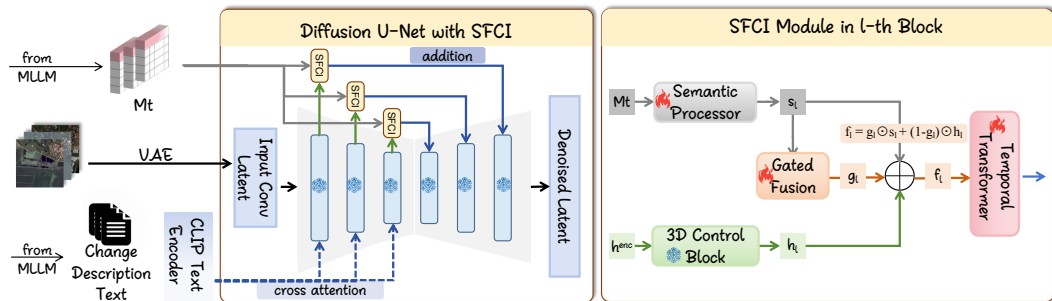

Figure 4: The **Semantic-Fused Control Injection (SFCI)** mechanism. SFCI guides the diffusion U-Net by dynamically fusing signals from two parallel streams: a **Structural Path** (green) that extracts low-level spatio-temporal dynamics, and a **Semantic Path** (gray) that translates the MLLM's high-level temporal understanding ($\mathbf{M}_t$). The resulting fused, multi-scale control signal is injected into the U-Net decoder to ensure temporally consistent forecasting.

injected into the corresponding U-Net decoder layer by adding it to the skip connection features, which allows TAMMs to modulate the generation process at multiple spatial scales, ensuring the final forecast image aligns with both the low-level structural evolution and the high-level semantic changes derived from the MLLM.

**Staged Training Strategy.** We employ a two-stage training strategy to stably integrate semantic guidance. First, we train only the structural path to learn foundational spatio-temporal priors from the image data. Then, with the structural path frozen, we train the semantic components to translate the MLLM's guidance into control signals, which prevents catastrophic forgetting. Both stages optimize a standard diffusion loss. A detailed breakdown of the training protocol and loss formulations is provided in Appendix A.6.

## 4 EXPERIMENTS

### 4.1 EXPERIMENTAL SETUP

#### 4.1.1 DATASETS

Our experiments leverage a large-scale dataset curated for temporal analysis. The training set consists of 37,003 multi-temporal image sequences from the fMoW dataset Christie et al. (2018), with descriptive labels automatically generated by Qwen2.5-VL model to enable training at scale. For rigorous and unbiased evaluation, we curated an independent testset of 150 challenging, long-term (5-10 years) sequences from Google Earth. The ground-truth change descriptions for testset were annotated by human experts, providing a high-quality benchmark for evaluation. Further details on our data curation process, annotation prompts, and data distribution are provided in Appendix A.7.

#### 4.1.2 EVALUATION METRICS.

We evaluate our model's performance on two tasks. For assessing the textual descriptions summarizing temporal changes, we utilize standard natural language generation metrics: BLEU-4 Papineni et al. (2002), METEOR Lavie & Agarwal (2007), ROUGE-L Lin (2004), and CIDEr-D Vedantam et al. (2015), which collectively measure aspects like n-gram precision, word alignment considering synonyms, longest common subsequence overlap, and consensus among reference descriptions. To evaluate predicted images, besides standard metrics PSNR Zhang et al. (2020), SSIM Wang et al. (2004), and LPIPS Zhang et al. (2018), we further introduce TCS metric for evaluating temporal image generation, which has detailed in Section 2.3.

#### 4.1.3 IMPLEMENTATION DETAILS

Our framework is conducted on two NVIDIA A6000 GPUs. The MLLM backbone is a frozen DeepSeek-VL2 Wu et al. (2024) due to its MoE architecture and open-source accessibility, while the generative component adapts the U-Net architecture from DiffusionSat Khanna et al. (2024) which is based on Stable Diffusion 2-1. We initially considered native unified models (Chameleon Team

| Model | Temporal Change Description | | | | Future Satellite Image Forecasting | | | |
|---|---|---|---|---|---|---|---|---|
| | B-4↑ | M↑ | R-L↑ | C-D↑ | P↑ | S↑ | L↓ | TCS↑ |
| *Baselines for Temporal Change Description* | | | | | | | | |
| RSICC-Former Liu et al. (2022) | 0.1285 | 0.1930 | 0.3489 | 0.5344 | - | - | - | - |
| SITSCC Peng et al. (2024) | 0.2122 | 0.2961 | 0.4701 | 0.6244 | - | - | - | - |
| Qwen2.5-VL-7B Bai et al. (2025) | 0.2089 | 0.2845 | 0.4612 | 0.7156 | - | - | - | - |
| DeepSeek-VL2 Wu et al. (2024) | 0.1920 | 0.2382 | 0.4603 | 0.7362 | - | - | - | - |
| TEOChat Irvin et al. (2025) | 0.2398 | 0.3102 | **0.4735** | 0.8267 | - | - | - | - |
| *Baselines for Future Satellite Image Forecasting* | | | | | | | | |
| Geosynth-Canny Sastry et al. (2024) | - | - | - | - | 11.3752 | 0.1757 | 0.7559 | 0.2170 |
| MCVD Voleti et al. (2022) | - | - | - | - | 9.2208 | **0.2098** | 0.4970 | 0.1930 |
| DiffusionSat Khanna et al. (2024) | - | - | - | - | 11.8878 | 0.1520 | 0.5225 | 0.7624 |
| *Our Unified Model* | | | | | | | | |
| TAMMs (Ours) | **0.2669** | **0.3312** | 0.4690 | **0.9030** | **12.0697** | 0.1831 | **0.4931** | **0.9690** |

Table 1: Comprehensive evaluation on Temporal Change Description and Future Satellite Image Forecasting tasks. Metrics are abbreviated: B-4 (BLEU-4), M (METEOR), R-L (ROUGE-L), C-D (CIDEr-D), P (PSNR), S (SSIM), L (LPIPS). Models are grouped by their primary task. A dash (-) indicates the metric is not applicable for that model. **Bold** is best; underline is second best.

(2024), Janus-Pro Chen et al. (2025)) which could theoretically leverage cross-task information more effectively, but preliminary experiments showed significant performance gaps compared to specialized models, making them unsuitable for demanding remote sensing tasks.

A critical component of our methodology is a multi-stage training strategy for the generative module. This approach is designed to first establish robust spatio-temporal structural control before integrating the MLLM's high-level semantic guidance. This decoupling prevents catastrophic forgetting and ensures a stable fusion of the low-level structural priors and high-level semantic reasoning. A detailed breakdown of our three-stage training protocol, including specific learning rates and optimizer schedules for each stage, is provided in Appendix A.9.2. Comprehensive architectural details and a full list of hyperparameters can also be found in Table 5.

## 4.2 METHOD COMPARISON

### 4.2.1 BASELINES

We evaluate TAMMs against comprehensive set of state-of-the-art methods for two primary tasks.

**For Temporal Change Description,** we compare against three categories of models: (1) Specialized remote sensing change captioning models, including the bi-temporal RSICC-Former Liu et al. (2022) and the multi-temporal SITSCC Peng et al. (2024). (2) General-purpose MLLMs, including Qwen2.5-VL-7B Bai et al. (2025) and DeepSeek-VL2 Wu et al. (2024), evaluated in finetuned settings on same datasets as TAMMs to assess their native temporal reasoning abilities. (3) A specialized temporal MLLM, TEOChat Irvin et al. (2025), designed for earth observation conversations.

**For Future Satellite Image Prediction,** our baselines represent three distinct conditioning strategies: (1) Methods relying on single-frame structural guidance, represented by GeoSynth-Canny Sastry et al. (2024), which uses edge maps from only the last observed image. (2) Approaches adapted from video prediction, represented by MCVD Voleti et al. (2022), a conditional diffusion model. (3) Models using metadata conditioning, represented by DiffusionSat Khanna et al. (2024), a generative foundation model for satellite imagery.

### 4.2.2 QUANTITATIVE RESULTS

Table 1 shows that TAMMs demonstrates superior resultscompared to all evaluated baselines.

In temporal change description, TAMMs outperforms prior methods such as SITSCC (0.2669 vs. 0.2122 BLEU-4, 0.3312 vs. 0.2961 METEOR) and DeepSeek-VL2 (0.9030 vs. 0.7362 CIDEr-D), highlighting its enhanced temporal reasoning and generation fidelity. For future satellite image prediction, TAMMs achieves the highest PSNR (12.07), competitive SSIM (0.1831 vs. 0.2098 for MCVD), and the best LPIPS (0.4931).

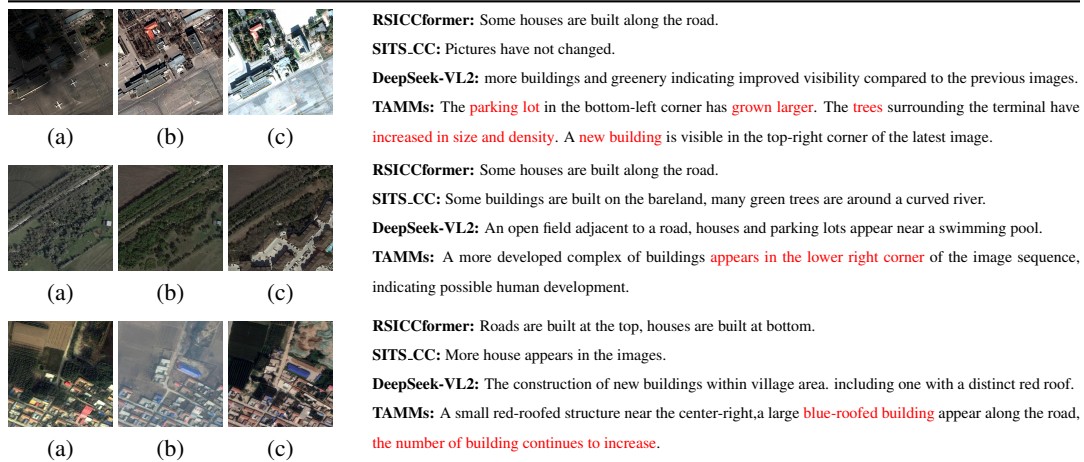

Figure 5: Qualitative comparison of temporal change description methods. Sequential images (a-c) are shown for each example, followed by outputs from baseline models and TAMMs. It can be seen that TAMMs has noticed many change details that other models have not paid attention to.

We notice the improvements in standard quality metrics (SSIM) are modest, this pattern is consistent with recent diffusion-based forecasting literature, where models that better preserve high-frequency and structural details often trade a small amount of PSNR/SSIM for more realistic and temporally consistent predictions Khanna et al. (2024). In our setting, such perceptual–temporal fidelity (captured by LPIPS and especially TCS) is more critical than maximizing distortion-based scores, because downstream change analysis depends on plausible evolution rather than exact pixel-wise reconstruction. TAMMs demonstrates a substantial advantage in TCS, achieving 0.9690, compared to 0.7624 for DiffusionSat and only 0.2170 for GeoSynth-Canny. This TCS margin indicates TAMMs' superior ability to produce temporally coherent futures grounded in prior dynamics.

These quantitative gains are visually reflected in Figures 5 and 6, where TAMMs yields more faithful change summaries and future predictions with better temporal structure.

## 4.3 ABLATION STUDIES

While above results demonstrate overall superiority of TAMMs, to understand the specific contributions of key architectural innovations, we conducted a series of ablation studies, with results presented in Table 2, confirms each proposed module is essential for the framework's performance.

For the temporal understanding task, removing PTE and CTP modules leads to significant performance degradation, most notably dropping TCS score by 23% and 15% respectively. This demonstrates their critical role in awakening MLLM's ability to comprehend temporal dynamics accurately. For future forecasting task, results highlight the importance of our MLLM-guided approach. Disabling semantic feature fusion ('w/o Semantic Fusion') causes a substantial 18% drop in TCS (from 0.9690 to 0.7911), more severe impact than removing only high-level text guidance. This finding validates our core hypothesis: injecting the MLLM's deep semantic reasoning directly into the generative control path is paramount for ensuring long-term temporal consistency. The 'Base Control Block' variant, which lacks both forms of semantic guidance, performs the worst, confirming that all components are complementary and necessary for achieving state-of-the-art performance.

To ensure our evaluation's robustness, we analyzed the average length of the generated descriptions. As detailed in Appendix A.8 , TAMMs is the only model whose output length closely aligns with the ground-truth references , confirming its superior metric scores are not artifact of generation length.

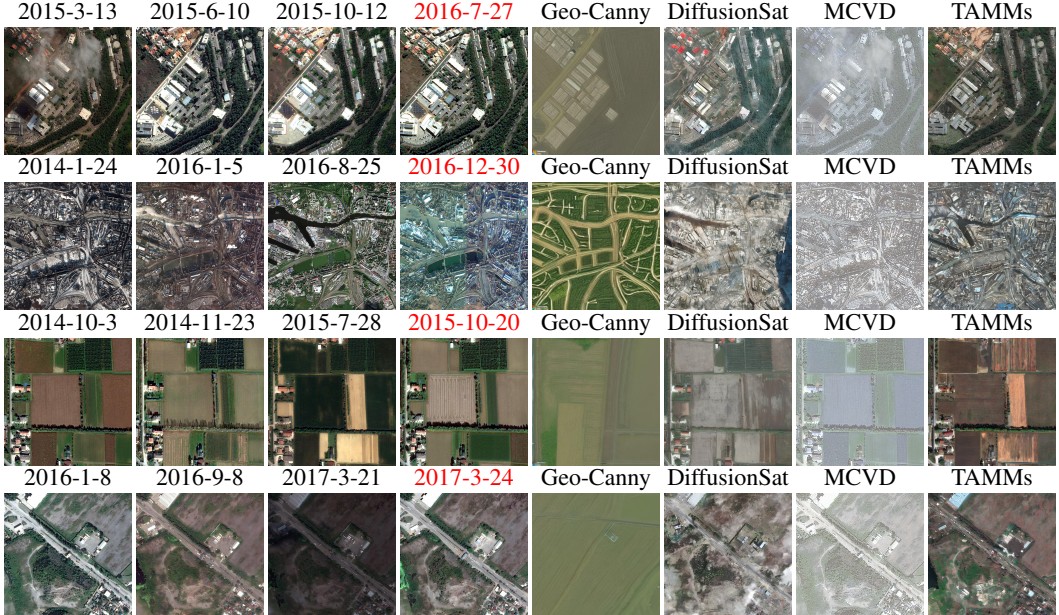

Figure 6: Generated samples from our temporal prediction task. Left columns are input sequences and target (with red dates indicating the target images to be predicted); right are predictions from different models. It can be clearly seen that TAMMs well captures the changing trends in the sequence and also ensures good image quality.

| Model / Config | Change Description Metrics | | | | Image Prediction Metrics | | | |
|---|---|---|---|---|---|---|---|---|
| | B-4↑ | M↑ | R-L↑ | C-D↑ | P↑ | S↑ | L↓ | TCS↑ |
| *Ablation on Temporal Adaptation (for Change Description)* | | | | | | | | |
| SFT only | 0.2134 | 0.2901 | 0.4634 | 0.7523 | - | - | - | 0.6842 |
| w/o PTE | 0.2387 | 0.3089 | **0.4721** | 0.8234 | - | - | - | 0.7456 |
| w/o CTP | 0.2445 | 0.3156 | 0.4678 | 0.8567 | - | - | - | 0.8234 |
| *Ablation on MLLM Integration (for Image Prediction)* | | | | | | | | |
| w/o Semantic Fusion | - | - | - | - | 12.0642 | 0.1596 | 0.5198 | 0.7911 |
| w/o Text Guidance | - | - | - | - | 11.9655 | 0.1709 | 0.5065 | 0.9410 |
| Base Control Block | - | - | - | - | 11.8878 | 0.1520 | 0.5225 | 0.7624 |
| *Our Full Model* | | | | | | | | |
| **TAMMs (Full)** | **0.2669** | **0.3312** | 0.4690 | **0.9030** | **12.0697** | **0.1831** | **0.4931** | **0.9690** |

Table 2: Ablation studies for both tasks.

# 5 RELATED WORKS

## 5.1 MULTIMODAL LARGE LANGUAGE MODELS IN REMOTE SENSING

Multimodal Large Language Models (MLLMs) have shown exceptional capabilities in integrating vision and language Wang et al. (2024b;a). However, their application to SITS is non-trivial. General-purpose models, even those with video understanding like Qwen2.5-VL Bai et al. (2025), are not optimized for the sparse, long-term nature of SITS data Khanna et al. (2024). While satellite-specific adaptations like SITSCC Peng et al. (2024) and Time-VLM Zhong et al. (2025) have emerged for change captioning, they typically rely on handcrafted temporal modules and, crucially, are limited to descriptive tasks without generative forecasting capabilities.

## 5.2 CONTROLLABLE GENERATION FOR FUTURE FORECASTING

Latent Diffusion Models (LDMs) Rombach et al. (2022), particularly when combined with spatial control mechanisms like ControlNet Zhang et al. (2023), have set the state of the art in high-fidelity image synthesis. In remote sensing, models like DiffusionSat Khanna et al. (2024) have adapted this

technology using metadata conditioning. The fundamental limitation, however, is that their control signals are typically low-level (e.g., edge maps) or based on simple metadata, lacking guidance from a high-level, semantic understanding of an evolving temporal narrative. They can generate what a scene 'looks like', but cannot reason about 'why' it should look that way based on historical changes.

## 6 CONCLUSION AND FUTURE WORK

In this paper, we addressed the fragmented nature of SITS, where temporal change description (TCD) and future forecasting (FSIF) are treated as disjointed tasks limited by a common bottleneck in long-range temporal understanding. We proposed **TAMMs**, the first unified framework designed to synergistically solve both, built on core hypothesis that empowering MLLM to deeply comprehend historical dynamics would enable more consistent future forecasting. Our comprehensive experiments validate that TAMMs achieves state-of-the-art performance on both tasks, significantly outperforming specialist baselines. Critically, its substantial gains on our proposed **TCS** metric provide strong evidence that unified approach leads to forecasts are demonstrably more consistent with historical evolution, establishing a new paradigm for reasoning-based SITS analysis.

Building on this work, several promising research avenues are identified. To enhance forecasting over very long-term horizons and for rare, abrupt events, future work will explore hierarchical temporal modeling and the integration of external data sources. To improve the framework's efficiency and reliability for real-world applications, we will also investigate model compression techniques and incorporate uncertainty quantification (e.g., through Bayesian approaches) to move beyond deterministic predictions and provide richer forecasting outputs.

## ACKNOWLEDGEMENTS

This work was supported by the Foundation Program of the Key Laboratory of Science and Technology on Complex Electronic System Simulation under Grant 614201001032203. We are grateful to our colleagues and research experts in remote sensing and aerospace for their constructive discussions and guidance throughout this work.

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

## A    APPENDIX

### A.1    ETHICS STATEMENT

Our work strictly follows the ICLR Code of Ethics. This study was conducted without involving any human subjects or animal experimentation. We have taken great care to ensure the ethical use of all datasets, including Fmow Christie et al. (2018), by strictly adhering to their official usage guidelines. Our research process was designed to prevent any privacy breaches or security vulnerabilities, and no personally identifiable information was used at any point. Furthermore, we are committed to actively addressing potential biases and avoiding discriminatory outcomes. This research is a testament to our commitment to integrity and transparency.

### A.2    REPRODUCIBILITY STATEMENT

We have made every effort to ensure our results are fully reproducible. This paper provides a detailed description of our experimental methodology, including training steps, model configurations, and hardware specifications. The key component of our work, the TAMMs, is also thoroughly described to enable replication. All datasets used, such as the publicly available Fmow Christie et al. (2018) dataset, are accessible to the public, ensuring consistent evaluation. We believe these measures provide other researchers with the necessary information to reproduce our findings and build upon our work.

### A.3    LLM USAGE

In preparing this manuscript, we employed a Large Language Model (LLM) as a supportive writing tool. The LLM's role was confined to assisting with textual refinement, focusing on improving the clarity, readability, and linguistic quality of the paper. This included tasks such as rephrasing for better flow and correcting grammatical errors.

We want to be explicit that the LLM was not used for any part of the core scientific process. It did not contribute to the research's conceptual framework, the experimental methodology, or the analysis

and interpretation of results. All scientific content, ideas, and conclusions are the sole responsibility of the authors. By transparently stating our use of the LLM, we commit to a high standard of research integrity. We have thoroughly vetted all LLM-assisted text to ensure its accuracy and originality, and we take full accountability for all content presented in this paper.

### A.4 CONTEXTUAL TEMPORAL PROMPT

```
<image>...<image> Scene: [Scene Description]. Describe
specific changes between these time-series remote sensing
images in a single paragraph. Focus on concrete changes to
structures, landscape, or development with precise location
details.
```

### A.5 PROMPT ROBUSTNESS ANALYSIS DETAILS

To evaluate the robustness of our Contextual Temporal Prompting (CTP), we conducted an additional ablation study using three variants of prompts with varying levels of detail to test the stability of our model:

- $\mathcal{P}_{Short}$: "Describe the changes in the satellite images." (Minimal instruction).
- $\mathcal{P}_{Ours}$ (Default): The structured prompt used in the paper (see Appendix A.4).
- $\mathcal{P}_{Verbose}$: A complex prompt adding approximately 50 words of context about specific land-cover types.

The full text for the $\mathcal{P}_{Verbose}$ prompt is as follows:

```
<image>...<image> Scene: [Scene Description]. Analyze
the provided satellite image time series with particular
attention to diverse land-cover categories, including
urban infrastructure, agricultural vegetation patterns, and
hydrological features. Please distinguish between transient
seasonal effects (e.g., phenological changes) and permanent
structural developments. Based on this context, describe
specific changes between these time-series remote sensing
images in a single paragraph. Focus on concrete changes to
structures, landscape, or development with precise location
details.
```

The quantitative results are presented in Table 3. As shown, the performance variance is minimal (TCS fluctuates $< 0.015$). This demonstrates that TAM is robust: the learnable visual alignment plays the primary role, while the prompt serves as a high-level task trigger.

Table 3: Prompt Robustness Analysis.

| Prompt Type | TCS | BLEU-4 | CIDEr-D |
|---|---|---|---|
| $\mathcal{P}_{Short}$ | 0.9580 | 0.2610 | 0.8910 |
| $\mathcal{P}_{Ours}$ (Default) | **0.9690** | 0.2669 | **0.9030** |
| $\mathcal{P}_{Verbose}$ | 0.9615 | **0.2675** | 0.8980 |

### A.6 TRAINING STRATEGY FOR THE GENERATIVE MODULE

Training the generative component of TAMMs, particularly the Enhanced Control Module (ECM), requires a carefully designed strategy to ensure stable learning and effective fusion of structural and semantic information. A naive joint training approach risks suboptimal performance, as the model may struggle to learn low-level spatio-temporal dynamics while simultaneously trying to interpret high-level semantic guidance. To address this, we propose a **two-stage training procedure**

designed to decouple these learning objectives, prevent catastrophic forgetting of structural priors, and facilitate a robust integration of semantic control.

The trainable parameters of the ECM are partitioned into two sets: the structural components $\theta_{st}$, which comprise the 3D Control Blocks responsible for capturing physical spatio-temporal patterns, and the semantic-temporal components $\theta_{s-t}$, which include the Semantic Processors, Gated Fusion units, and Temporal Transformers responsible for translating the MLLM's reasoning into guidance.

### A.6.1 STAGE 1: STRUCTURAL CONTROL PRE-TRAINING

**Objective:** The primary goal of this stage is to teach the model the fundamental dynamics of spatio-temporal evolution based purely on the visual information within the SITS sequence. This stage establishes a robust structural foundation for the model.

**Setup:** We freeze the parameters of the pre-trained LDM U-Net ($\theta_{U\text{-}Net}$) and the entire MLLM module. Only the structural parameters $\theta_{st}$ of the ECM are set to be trainable. The semantic path of the ECM is bypassed by zeroing out its output, meaning the conditioning signal injected into the U-Net decoder is derived solely from the structural path ($\mathbf{z}'_l$ is derived from $\mathbf{h}_l^{(ctrl)}$).

**Loss Formulation:** Training minimizes the standard LDM diffusion loss. The noise prediction network $\epsilon_\theta$ is conditioned only on the noisy latent $z_\tau$, the timestep $\tau$, and the U-Net's internal encoder features $\{\mathbf{h}_l^{(enc)}\}$. The loss is formulated as:

$$\mathcal{L}_{\text{Stage 1}} = \mathbb{E}_{\mathbf{x}_0, \epsilon \sim \mathcal{N}(0,\mathbf{I}), \tau} \left[ \left\| \epsilon - \epsilon_\theta(z_\tau, \tau \mid \{\mathbf{h}_l^{(enc)}\}; \theta_{U\text{-}Net}^{\text{frozen}}, \theta_{st}^{\text{train}}) \right\|_2^2 \right] \tag{9}$$

By backpropagating only through $\theta_{st}$, this stage effectively pre-trains the 3D Control Blocks to extract meaningful and coherent temporal structures from the U-Net's latent space.

### A.6.2 STAGE 2: SEMANTIC AND TEMPORAL REFINEMENT

**Objective:** With a strong structural prior established, the goal of this stage is to learn how to effectively translate the MLLM's high-level semantic understanding ($\mathbf{M}_t$) into fine-grained control signals and adaptively fuse them with the pre-trained structural features.

**Setup:** We keep the LDM U-Net ($\theta_{U\text{-}Net}$) and the now pre-trained structural parameters ($\theta_{st}$) frozen. The MLLM module is active (but also frozen) to provide the semantic context vector $\mathbf{M}_t$. Only the semantic-temporal parameters $\theta_{s-t}$ are set to be trainable. The full SFCI pathway, including the adaptive gated fusion, is now active.

**Loss Formulation:** Training again minimizes the diffusion loss, but the noise prediction network is now additionally conditioned on the semantic vector $\mathbf{M}_t$. The gradients are computed only with respect to $\theta_{s-t}$:

$$\mathcal{L}_{\text{Stage 2}} = \mathbb{E}_{\mathbf{x}_0, \mathbf{M}_t, \epsilon \sim \mathcal{N}(0,\mathbf{I}), \tau} \left[ \left\| \epsilon - \epsilon_\theta \left( z_\tau, \tau \mid \{\mathbf{h}_l^{(enc)}, \mathbf{M}_t\}; \theta_{U\text{-}Net}^{\text{frozen}}, \theta_{st}^{\text{frozen}}, \theta_{s-t}^{\text{train}} \right) \right\|_2^2 \right] \tag{10}$$

This staged approach is critical. By freezing the structural path, we prevent the powerful semantic signals from disrupting the already learned, delicate spatio-temporal priors. The adaptive gating mechanism (Eq. 5) can then focus solely on learning the optimal balance between these two information streams, leading to a stable and effective integration of high-level reasoning into the generative process.

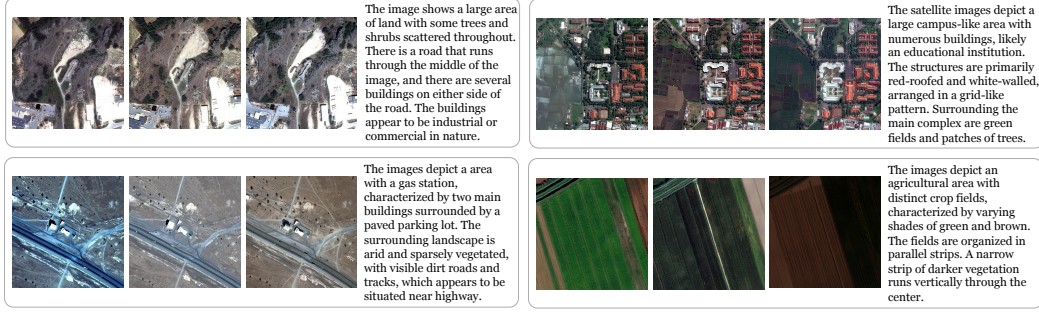

Figure 7: Samples of Satellite Image Time Series captions TAMMs used.

## A.7 DATASET DETAILS

### A.7.1 DATASET GENERATING PROMPTS

---

**Prompt for Scene Description Generation**

You are given a set of remote sensing images of the **same geographical area** captured at **different time points**. Your task is to produce a **concise but detailed description** of the **scene itself**, based on the entire set of images. Do **not** describe any changes or temporal differences across the images.

Please follow these specific instructions:

1. **Objective**: Describe the physical and functional characteristics of the area shown across the entire image set **within 50 words**. Treat the images as a unified snapshot of a single location.

2. **Content Scope**: Your description should integrate and reflect **as much specific and fine-grained detail as possible**, including but not limited to:

    - **Natural features** (e.g., rivers, forests, vegetation patterns, terrain types)
    - **Artificial structures** (e.g., buildings, roads, bridges, industrial zones, residential areas)
    - **Spatial layout and distribution** (e.g., clustered housing, grid-like road networks, green spaces between zones)
    - **Geographical context** (e.g., inland plain, coastal region, mountainous area)
    - **Probable functional uses or land usage** (e.g., commercial, agricultural, recreational, mixed-use)

3. **Writing Style**:

    - Maintain a **scientific and objective tone**. Avoid speculation unless supported by clear visual evidence.
    - Emphasize **spatial relationships** and **layout patterns** when relevant (e.g., "A dense network of roads connects uniformly spaced residential blocks").
    - Use **precise, domain-appropriate language** to ensure clarity and specificity.
    - Avoid vague adjectives (e.g., "nice," "big") in favor of **quantifiable or structural descriptors**.

4. **Exclusion Note**: You must **not** mention or infer any **temporal changes**, differences between time points, or dynamic aspects. The output should reflect only the **static characteristics** of the observed area.

---

---

**Prompt for Temporal Change Description Generation**

**Scene:** {category}. You are looking at a **time series of remote sensing images** captured over the **same geographical location** across multiple time points.
Your task is to generate a description, in **English**, that highlights and summarizes **only the specific changes** observed from the **earliest** to the **latest** image. Your output must be:

1. **Length Constraint**: Strictly limited to **under 50 words** (even though initial analysis may consider 77). Be as concise and informative as possible.

2. **Focus on Changes Only**:
   - Do **not** describe the general scene or static elements.
   - Report **concrete, observable changes** over time.

3. **Types of Changes to Identify**:
   - **Structural developments**: newly built or demolished buildings, roads, runways, industrial facilities, housing blocks.
   - **Landscape transformations**: deforestation, vegetation expansion/reduction, coastline shifts, mining excavation.
   - **Functional conversions**: farmland turned into residential zone, industrial zones expanding into greenfields, infrastructure upgrades.

4. **Style and Precision**:
   - Use **precise spatial references** (e.g., "new building in bottom-left corner," "road extended eastward," "tree cover lost near central canal").
   - Maintain a **factual, scientific, and neutral tone**—no speculation or subjective descriptions.
   - Use **clear, domain-appropriate vocabulary** for spatial and structural changes.

**Reminder:** Your output should clearly reflect the **differences over time**, not general characteristics of the area. Prioritize meaningful, observable developments with accurate location cues.

---

### A.7.2 SCENE CAPTIONS SAMPLES

Figure 7 presents samples of Satellite Image Time Series captions generated by Qwen2.5-VL Bai et al. (2025).

### A.7.3 DATASET DISTRIBUTION

The dataset utilized for training TAMMs is meticulously curated from the large-scale FmoW benchmark. Recognizing the specific demands of temporal change understanding and prediction tasks, we implemented a selection process targeting high-quality, multi-temporal sequences. This process involved filtering for sequences with sufficient temporal coverage (e.g., minimum number of distinct timestamps) and exhibiting discernible changes relevant to common land use and environmental dynamics.

The resulting curated dataset comprises 37003 high-fidelity temporal sequences, representing 58 distinct and fine-grained semantic categories. These categories encompass a diverse range of real-world scenarios, including detailed infrastructure developments (e.g., *airport terminal expansion*, *solar farm construction*), agricultural shifts (e.g., *crop field rotation*, *irrigation system changes*), and residential area evolution, ensuring a robust testbed for fine-grained temporal analysis.

Figure 8 provides a visual summary of the category distribution within this refined dataset. The visualization employs a dual-axis format: the bars (left Y-axis) depict the absolute number of sequences per category, while the line plot (right Y-axis) shows the corresponding percentage distribution. Color coding distinguishes categories based on their relative frequency within our curated set. While some long-tail characteristics persist, reflecting real-world data distribution, our filtering ensures that the dataset primarily consists of clear, high-quality examples suitable for training and

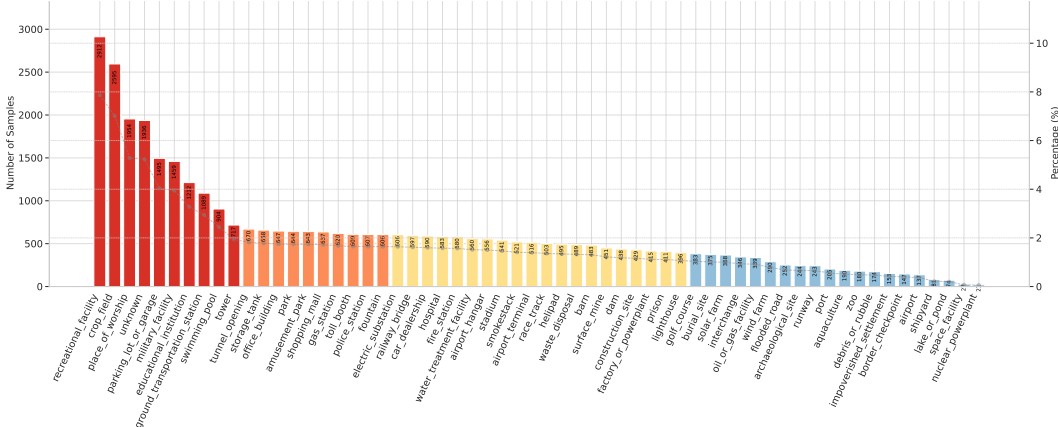

Figure 8: Category distribution in the remote sensing dataset. The bar chart (left Y-axis) shows the sample counts per category. A dual axis line plot (right Y-axis) overlays the relative percentage each category contributes to the total dataset. Categories are sorted in descending order of frequency. Colors encode category rank: red (top 10), orange (top 20), yellow (mid range), blue (long tail).

evaluating sophisticated temporal-aware models that require fine-grained understanding of changes over time.

### A.8 CONCRETE GENERATED TEXT LENGTH ANALYSIS

To address potential concerns about the sensitivity of automatic evaluation metrics (e.g., BLEU, CIDEr-D) to the length of generated text, we conducted a comprehensive analysis of the average word count for all generated descriptions compared to the human-annotated ground truth references on our test set. The results are presented in Table 4.

| Model | Ave Length (words) | $\Delta$ from GT (words) |
|---|---|---|
| Ground Truth | 30.89 | 0.00 |
| RSICC-Former | 15.20 | 15.69 |
| SITSCC | 19.03 | 11.86 |
| DeepSeek-VL2 | 34.55 | 3.66 |
| **TAMMs (Ours)** | **32.23** | **1.34** |

Table 4: Average word count analysis. The '$\Delta$ from GT' column shows the absolute difference from the Ground Truth length, highlighting that TAMMs' output length is most aligned with human references.

The analysis reveals significant insights into the quality and nature of the descriptions produced by different models.

- **Under-generation by Baselines:** Prior specialized methods like RSICC-Former (15.20 words) and SITSCC (19.03 words) produce descriptions that are substantially shorter than the ground truth average of 30.89 words. Their length difference of 15.69 and 11.86 words, respectively, indicates a significant inability to capture the full complexity and scope of the temporal changes present in the satellite imagery sequences.

- **Over-generation by General MLLM:** In contrast, the general-purpose DeepSeek-VL2 model generates slightly longer descriptions (34.55 words) than the ground truth. While closer in length, its lower scores on semantic quality metrics (Table 1) suggest that this extra length does not necessarily translate to higher quality, potentially indicating verbosity or less relevant details.

- **Optimal Alignment by TAMMs:** Our model is the only that produces descriptions with an average length (32.23 words) that is remarkably close to the ground truth, with a minimal difference of only 1.34 words. This close alignment strongly supports the conclusion that our model's superior

performance in automatic metrics stems from genuinely higher-quality generation that comprehensively and concisely captures the key temporal changes, rather than from any length-related bias in the evaluation metrics. TAMMs achieves the optimal balance of comprehensive temporal coverage while maintaining the appropriate descriptive scope.

## A.9 TRAINING DETAILS

### A.9.1 MODEL ARCHITECTURE AND HYPERPARAMETERS

We adopt a U-Net architecture based on DiffusionSat for the core diffusion model, enhanced with our MLLM-based temporal understanding modules (PTE, CTP) and the Semantic-Fused Control Injection (SFCI) mechanism via the Enhanced ControlNet Module. Key architectural choices and default hyperparameters used during training are summarized in Table 5. Note that specific learning rate schedules varied across training stages as detailed below.

### A.9.2 MULTI-STAGE TRAINING STRATEGY

The training process for our Enhanced Control Module (ECM) follows a carefully designed three-stage strategy. This is crucial to effectively integrate semantic guidance from the MLLM while preserving the foundational structural control learned by the base 3D Control Blocks. The stages below focus specifically on optimizing the diffusion loss ($\mathcal{L}_{\text{diff}} = \mathbb{E}\left[\|\epsilon - \epsilon_\theta(\mathbf{z}_\tau, \tau, \text{cond})\|_2^2\right]$) for the Enhanced ControlNet Module and ensuring stable integration:

**Stage 1: Structural Control Pre-training.** The initial stage focuses on learning the foundational spatio-temporal dynamics. We freeze the U-Net and MLLM, training only the 3D Control Block weights ($\theta_{st}$). This allows the model to learn to extract structural priors from the image sequence alone, without semantic influence. This stage was trained for approximately 50k steps using the AdamW optimizer Kingma & Ba (2014) with a constant learning rate of $\texttt{2e-5}$.

**Stage 2: Initial Semantic Fusion Training.** With the structural path frozen, we then introduce the MLLM's semantic guidance ($\mathbf{M}_t$). In this stage, we train only the semantic path components ($\theta_{s-t}$), including the Semantic Processors and Gated Fusion modules. This teaches the model how to translate the MLLM's abstract understanding into features compatible with the generative process. This stage also used a constant learning rate of $\texttt{2e-5}$.

**Stage 3: Joint Fine-tuning with Cosine Decay.** Finally, we fine-tune the semantic path components ($\theta_{s-t}$) to achieve optimal convergence and stability. The key difference in this stage is the learning rate schedule, which is switched to a Cosine Annealing scheduler. This allows for a more refined integration of the semantic guidance with the now-robust structural priors, leading to the final model performance.

| Condensed Two-Stage Training Flow |
| --- |
| **Initialize:** Frozen LDM U-Net, Frozen MLLM, |
| **Trainable ECM components** ($\theta_{st}$, $\theta_{s-t}$)**.** |
| **Stage 1: Structural Control Learning** |
| **Goal:** Learn spatio-temporal structure from U-Net features. |
| **Active Components:** 3D Control Blocks ($\theta_{st}$). |
| **Inactive:** MLLM semantic input ($\mathbf{M}_t$), Semantic/Temporal ECM parts ($\theta_{s-t}$). |
| **Training:** Minimize LDM loss $\mathcal{L}_{\text{Stage 1}}$ w.r.t. **only** $\theta_{st}$. |
| **Stage 2: Semantic and Temporal Refinement** |
| **Goal:** Integrate MLLM guidance and refine temporal consistency. |
| **Active Components:** Full ECM path using frozen $\theta_{st}$ and MLLM context $\mathbf{M}_t$. |
| **Trained:** Semantic Processors, Gates, Temporal Transformers ($\theta_{s-t}$). |
| **Training:** Minimize LDM loss $\mathcal{L}_{\text{Stage 2}}$ w.r.t. **only** $\theta_{s-t}$. |
| **Output:** Trained ECM ($\theta_{st}$ from Stage 1, $\theta_{s-t}$ from Stage 2). |

### A.9.3 TRAINING INFRASTRUCTURE

All experiments were conducted on $2\times$NVIDIA A6000 GPUs using PyTorch 2.0.1. Mixed-precision training (AMP) was utilized to optimize memory usage and training speed. Training progress and metrics were logged using `wandb`.

### A.9.4 LOSS CURVES

Figure 9 illustrates the training loss progression. The top plot shows the loss associated with the temporal change description task (e.g., MLLM's text output loss). The bottom plot displays the diffusion loss ($\mathcal{L}_{\text{diff}}$) for the image generation component across the three training stages described in Section A.9.2. Stage 1 (approx. 50k steps, green) shows the initial convergence of the structural control path. Stage 2 (approx. 50k-70k steps, purple/pink) corresponds to the introduction and training of semantic fusion components with a constant learning rate. Stage 3 (approx. 70k+ steps, blue/teal) demonstrates the fine-tuning phase using a cosine learning rate decay, leading to further stabilization and refinement of the diffusion loss. The overall stable convergence across stages validates our training strategy.

| Parameter | Value |
|---|---|
| Epochs | 100 |
| Batch Size | 12 |
| Gradient Accumulation | 1 |
| Initial Learning Rate | `2e-5` |
| Optimizer | AdamW |
| Default Scheduler | Cosine Annealing (Stage 3) |
| Max Grad Norm | 1.0 |
| AMP (Mixed Precision) | Enabled |
| Workers | 12 |
| Model Max Length | 512 |
| Input Sequence Length | 3 |
| Prediction Length | 1 |
| Patch Size | 16 |
| Patch Stride | 8 |
| U-Net Encoder Dim | 1024 |
| Feedforward Dim | 2048 |
| Attention Heads | 8 |
| Dropout Rate | 0.1 |
| Visual Feature Dim | 1280 |
| Conditioning Dim (Diffusion) | 512 |
| Vision Mapping Layers | 2 |
| Text Feature Dim | 512 |
| Text Output Dim | 1024 |
| Fusion Dim | 1024 |
| Use Contrastive Loss | ✓(weight = 0.1) |
| Use MLLM Features | ✓ |
| Dual Modality Output | ✓ |
| Text Loss Enabled | ✓ |
| Output Attention Map | ✓ |

Table 5: Training configuration and architectural hyperparameters.

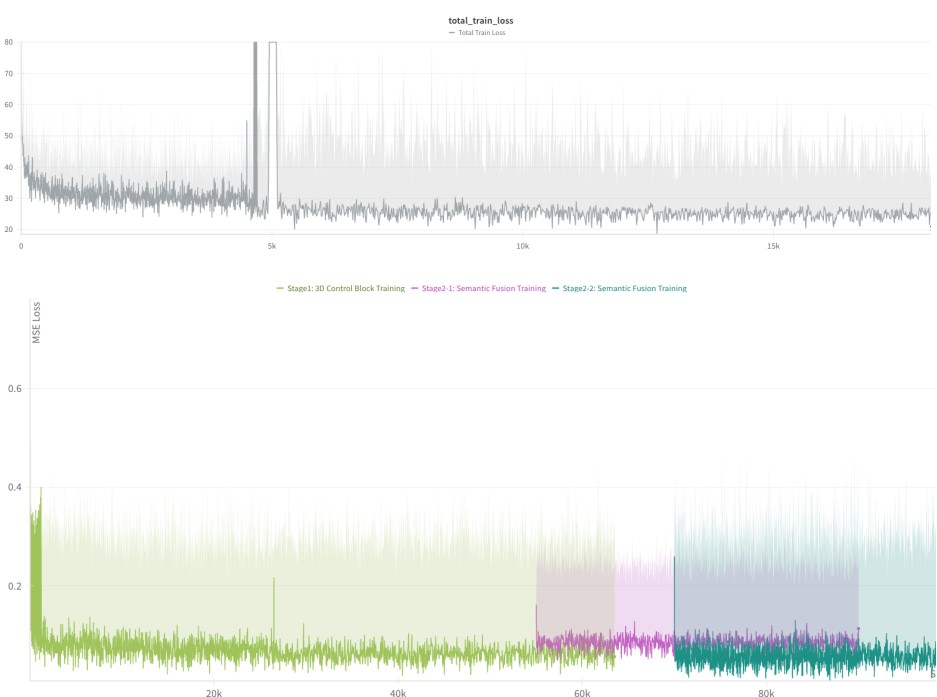

Figure 9: Training loss curves. (Top) Loss for the temporal change description task. (Bottom) Diffusion loss for the image generation component, illustrating the three training stages: Stage 1 (3D Control Block Pre-training, green region), Stage 2 (Semantic Fusion Training with constant LR, pink/purple region), and Stage 3 (Semantic Fusion Fine-tuning with cosine LR, blue/teal region). Steady convergence is observed in all stages.

