# OpenReview forum: "TAMMs: Change Understanding and Forecasting in Satellite Image Time Series with Temporal-Aware Multimodal Models"
_ICLR.cc/2026/Conference — ICLR 2026 Poster_

### Official Review · Reviewer_72WA · 2025-10-25

**Soundness:** 3
**Presentation:** 2
**Contribution:** 3
**Rating:** 6
**Confidence:** 4

**Summary:**

This paper presents TAMMs, the first unified framework that jointly performs temporal change description (TCD) and future satellite image forecasting (FSIF) by enhancing long-range temporal understanding. It introduces lightweight Temporal Adaptation Modules (PTE and CTP) to awaken a frozen MLLM’s ability to reason over sparse, long-term satellite image time series, producing a semantic feature Mt. The novel Semantic-Fused Control Injection (SFCI) mechanism then translates Mt into multi-scale, spatially-aware control signals to guide a frozen diffusion U-Net, ensuring patch-level temporal consistency in generated future images. Additionally, a new metric, Temporal Consistency Score (TCS), is proposed to quantify the plausibility of predicted changes against historical dynamics. Extensive experiments demonstrate that TAMMs significantly outperforms state-of-the-art specialized baselines on both tasks, with particularly large gains in TCS, validating the closed-loop paradigm of “understanding why changes occur to forecast what will happen.

**Strengths:**

TAMMs pioneers a single, frozen-backbone framework that couples satellite change captioning with future image synthesis, leveraging lightweight adapters to awaken long-range temporal reasoning and fine-grained diffusion control; it handles sparse, multi-year sequences, introduces the TCS metric to quantify temporal plausibility, and achieves comprehensive SOTA results with minimal trainable parameters.

**Weaknesses:**

1. the paper presentation need to be improved.
For example, TAMMs is designed through different modules like PTE, CTP, SFCI, however there are no corresponding label in figure 2, making it not convenient for readers to understand.
In addition, the descriptive labels are automatically generated by Qwen2.5-VL model, why in figure 2 it is ChatGPT icon?


2. The ablation study is not sufficient.
The choice of MLLM backbone(why DeepSeek-VL2) is not clear, as well as the choice of Stable Diffusion
In addition, in table2 Ablation on Temporal Adaptation (for Change Description), w/o PTE achieves sota and authors should provide corresponing analysis

**Questions:**

1.It is reasonable to encode time_diff into the input sequence, but why make it a learnable token, I do not think it is to“inject explicit temporal awareness” but implicit instead. The author should provide some persuasive ablation studies here.

2.In figure 5, apart from TAMMs generate the most detailed description, it is obvious that the description length of TAMMs is much longer than that of other models，  making it appear to be an unfair comparison, which is worth noting

---

> ### Author Response · Authors · 2025-11-24
> **Response to Reviewer 72WA**
>
> We are encouraged by the reviewer's assessment of TAMMs as a "pioneering" framework and the appreciation of our SOTA results. We address the constructive comments below.
>
> ### R# W1: Figure 2 Labels & Icons
>
> We appreciate the careful observation regarding the visualization. We clarify that the "ChatGPT" icon was merely a placeholder for a generic "Text Description" input and will be replaced by Qwen icon in the camera-ready version to avoid confusion. Furthermore, we will explicitly add distinct labels for the PTE, CTP, and SFCI modules directly into Figure 2. This revision will ensure the diagram strictly aligns with the methodology section, allowing readers to easily map the architectural components to their descriptions in the text.
>
> ### R# W2 & Q1: Learnable Token (PTE) & Ablation Analysis
>
> Regarding the necessity of the Physical Time Encoder (PTE), we respectfully argue that implicit learning from image sequences alone is insufficient for SITS. Unlike video data with fixed frame rates, satellite time series have highly irregular gaps ranging from days to years. A learnable MLP is essential to map these continuous time differences $\Delta t$ into a semantic space that the MLLM can interpret, distinguishing between short-term seasonal shifts and long-term structural evolution. This is empirically validated by our ablation study in **Table 2**, where removing the explicit learnable token ("w/o PTE") caused a significant drop in the CIDEr-D score from 0.9030 to 0.8234. This sharp performance degradation demonstrates that without the explicit, learnable time signal, the model struggles to reason correctly about the temporal magnitude of changes, confirming that the PTE is a critical, non-redundant component.
>
> Regarding the backbone choice, We selected DeepSeek-VL2 for its MoE architecture and open-source accessibility. Similarly, our SFCI module leverages DiffusionSat’s pre-training  which backbone uses the Stable Diffusion 2.1 architecture and shows good performance. We initially considered native unified models (Chameleon, Janus-Pro) which could theoretically leverage cross-task information more effectively, but preliminary experiments showed significant performance gaps compared to specialized models, making them unsuitable for demanding remote sensing tasks.
>
>
> ### R# Q2: Generated Text Length Fairness
>
> We respectfully point out that the comparison in Figure 5 is fair and actually highlights a key strength of our model: alignment with human ground truth. As already detailed in our quantitative analysis in **Appendix A.7 (Table 3)**, the average length of human expert annotations (Ground Truth) is **30.89 words**. TAMMs generates descriptions averaging **32.23 words** (a difference of only 1.34 words), whereas baselines like RSICC-Former average only **15.20 words**. The visual appearance of "longer description" for TAMMs simply reflects that our model successfully captures the detailed changes present in the Ground Truth, while baselines suffer from severe "under-generation" and miss critical semantic details. Therefore, the length difference is not an artifact of verbosity, but a measure of superior information recall.

---

### Official Review · Reviewer_kXkZ · 2025-10-29

**Soundness:** 3
**Presentation:** 1
**Contribution:** 2
**Rating:** 4
**Confidence:** 4

**Summary:**

This paper presents a framework called **TAMMs**, designed for temporal change description and future image forecasting in satellite image time series. The method builds upon a frozen MLLM and a diffusion-based generator, introducing two auxiliary modules — **Temporal Adaptation Module (TAM)** and **Semantic-Fused Control Injection (SFCI)** — to enable temporal awareness and semantic guidance without large-scale fine-tuning. The TAM module enhances the MLLM’s temporal reasoning through Physical Time Encoding (PTE) and Contextual Temporal Prompting (CTP), while the SFCI module maps the MLLM’s semantic outputs into multi-scale control signals to guide the diffusion generation process. The paper further proposes a new metric, **Temporal Consistency Score (TCS)**, to quantify the consistency between predicted changes and historical dynamics. Experiments conducted on the fMoW (training) and Google Earth (testing) datasets demonstrate that the proposed approach outperforms several existing baselines on both temporal description and image forecasting tasks.

**Strengths:**

1. The paper provides a unified framework for satellite image time-series modeling, integrating semantic understanding and future prediction into a single closed-loop process.
2. The proposed TAM and SFCI modules are logically designed to enable temporal encoding and generative control while keeping the pretrained MLLM and diffusion models frozen.
3. The introduction of the TCS metric offers a way to quantify spatial consistency between predicted and historical changes, addressing the limitation of conventional image quality metrics in temporal evaluation.

**Weaknesses:**

1. The study focuses narrowly on satellite image forecasting and does not explore the applicability of the proposed framework to other domains, limiting its generality.
2. The proposed approach appears primarily as an engineering implementation based on pretrained LLMs and diffusion models, with limited introduction of new theoretical mechanisms or modeling principles; thus, its novelty is modest.
3. The overall architecture is complex but lacks a clear explanation of the core ideas and hierarchical relationships among key components. The importance, dependency, and training connections of different modules are not well clarified. The paper allocates substantial space to implementation details and equations but does not adequately explain the design rationale or training phase distinctions. For example, while the appendix mentions multi-stage training, the main figures do not distinguish which components are frozen or trainable across different stages, which makes the workflow difficult to interpret.
4. Other issues: The title in the PDF and the OpenReview submission are inconsistent and should be unified; Equation (6) is missing a pair of braces.

**Questions:**

See Weaknesses.

---

> ### Author Response · Authors · 2025-11-24
> **Response to Reviewer kXkZ**
>
> We sincerely thank the reviewer for the time spent reviewing our work and for the detailed feedback. We value the opportunity to provide further context regarding our methodological contributions and the broader significance of the domain, which we hope will clarify the novelty of our approach.
>
> ### R# W1 & W2: Novelty & Domain Significance
>
> We appreciate the reviewer's perspective on the implementation and wish to elaborate on the algorithmic contributions of our framework. The core innovation of TAMMs is the **Semantic-Fused Control Injection (SFCI)** mechanism, which addresses a fundamental and non-trivial challenge in multimodal generation: effectively translating abstract, high-level semantic reasoning from an MLLM into fine-grained, multi-scale spatial control signals for diffusion models. This "Understanding-to-Forecasting" closed loop represents a methodological advancement beyond standard component integration, a design choice that other reviewers (e.g., R72WA and RzohA) have also recognized as "pioneering" and "well-motivated." Furthermore, regarding the domain scope, we emphasize that Satellite Image Time Series (SITS) analysis is not merely a niche application but a representative task for **long-range spatio-temporal reasoning** under sparse sampling conditions—a persistent challenge where general video models often fail. By solving these specific bottlenecks with a unified framework, our work offers insights applicable to broader temporal reasoning tasks beyond remote sensing.
>
> ### R# W3: Architecture Complexity & Training Strategy
>
> We apologize if the hierarchical relationship was not immediately clear in the initial reading. The complexity is a necessary result of our **decoupled multi-stage training strategy**, designed explicitly to ensure stability and prevent catastrophic forgetting. As detailed in **Appendix A.8.2**, the process is strictly structured: **Stage 1 (Structural)** freezes the MLLM/U-Net to train only the 3D Control Blocks (Green components) for physical dynamics; **Stage 2 (Semantic)** subsequently trains the Semantic Adapters (Pink components) to integrate the $M_t$ guidance. This design is not an arbitrary assembly but a deliberate approach to fuse high-level reasoning with low-level generation. To address your concern and improve readability, we will revise Figure 2 in the camera-ready version to explicitly color-code components according to their respective training stages, making the dependency and flow visually intuitive.
>
> ### R# W4: Typos & Title Consistency
>
> We thank the reviewer for the careful attention to detail. We have corrected the inconsistency between the PDF title and the submission metadata, and we have fixed the syntax error in Equation (6). These corrections will be fully reflected in the camera-ready revision.

---

### Official Review · Reviewer_7xU9 · 2025-10-31

**Soundness:** 2
**Presentation:** 2
**Contribution:** 2
**Rating:** 6
**Confidence:** 3

**Summary:**

The paper presents TAMMs, a unified framework for jointly performing Temporal Change Description (TCD) and Future Satellite Image Forecasting (FSIF) within a single MLLM-diffusion architecture. It introduces two main components: Temporal Adaptation Modules (TAM), which enhance the model’s ability to capture long-range temporal dynamics, and Semantic-Fused Control Injection (SFCI), which transfers temporal understanding into precise generative control. This integration allows insights from TCD to directly improve FSIF consistency and performance across both tasks.

**Strengths:**

- Synergistic training for change detection together with future generation is an interesting idea for enhancing the reasoning capabilities of MLLM in the satellite image domain.
- Temporal Consistency Score (TCS) adds value as a metric specifically desgned for application on SITS.
- Performance results are strong in Table 1.
- Ablation study is performed on proposed modules.

**Weaknesses:**

- Some examples of the data used for training (satellite images together with generated captions) should be presented, possibly in supp. material.
- writing could be improved, some non exhaustive examples for use of language include:
   - l.38 "because fails"
   - l.39-41 "the absence of a unified framework of temporal-aware multimodal model for satellite image change understanding and forecasting named TAMMs"
   - l.42 "How can model reason"
- similarly "Satellite Image Time Series (SITS)" appears in several places in text, one should be ok then use abbreviation

**Questions:**

- As far as i understand captions generated via Qwen 2.5 VL are treated as ground truths for training. Can you explain why the trained model outperforms Qwen 2.5?
- in l.199 can you explain what is meant via "specific time interval", what is the input to the small MLP?

---

> ### Author Response · Authors · 2025-11-24
> **Response to Reviewer 7xU9**
>
> We thank the reviewer for recognizing our "synergistic training" idea and the value of the TCS metric. We address the specific questions below.
>
> ### R# W1 & W2: Data Examples and Writing
>
> We will include representative training samples (image-caption pairs) in the revised Appendix to provide better context. We have also thoroughly proofread the manuscript to correct the language issues pointed out (e.g., "because fails", title consistency), ensuring a polished camera-ready version.
>
> ### R# Q1: Why outperform the teacher (Qwen2.5-VL)?
>
> We convince that this is a classic "Student outperforms Teacher" phenomenon in domain adaptation. While Qwen2.5-VL is a powerful generalist model, it is not optimized for the specific noise patterns and sparse temporal dynamics of SITS. By training lightweight adapters (TAM) specifically on SITS data, TAMMs filters out irrelevant visual noise and forces the model to align visual features strictly with *temporal change* narratives. This task specialization allows TAMMs to distinguish subtle remote sensing features (e.g., distinguishing construction progress from seasonal phenology) much more precisely than the zero-shot teacher model, which offers only generic descriptions.
>
> ### R# Q2: "Specific Time Interval" and MLP
>
> In Section 3.1, the "specific time interval" refers to the explicit duration $\Delta t_i$ (e.g., "365 days") between two consecutive images. The input to the MLP is the scalar numerical value of this time difference, which is then projected into a high-dimensional embedding and added to the `[TIME_DIFF]` token. This mechanism allows the MLLM to perform continuous temporal modeling by conditioning the attention mechanism on the magnitude of the time gap; for instance, it enables the model to understand that a small structural change might be negligible over 5 years but significant if it occurs over just 1 month.

---

### Official Review · Reviewer_zohA · 2025-11-01

**Soundness:** 3
**Presentation:** 3
**Contribution:** 3
**Rating:** 4
**Confidence:** 2

**Summary:**

The paper proposes TAMMs, a unified framework that connects TCD with FSIF. The approach freezes a multimodal LLM and introduces TAM for long-range temporal reasoning, and injects these semantics into a frozen diffusion U-Net via SFCI to guide fine-grained generation. A new metric, TCS, evaluates whether predicted changes align spatially and in area with historical change. Extensive experiments on a large fMoW training corpus and long-horizon Google Earth sequences show strong TCD performance and notable gains on TCS, with ablations supporting the contributions.
Strengths

**Strengths:**

1. Timely problem and clear unification, bridging understanding to forecasting in a single, parameter-efficient pipeline for remote-sensing time series.

2. Well-motivated architecture: TAM adds lightweight temporal conditioning without full fine-tuning; SFCI combines structural and semantic control for diffusion, yielding an intuitive, modular design.

**Weaknesses:**

1. On FSIF, some standard metrics are only competitive rather than strictly superior to strong generative baselines. A brief discussion and supplemental visuals would help reconcile this with the TCS gains.

2. Prompt robustness is missing: TAM relies on Contextual Temporal Prompting (CTP), but there is no analysis of robustness to prompt phrasing/length/templates; please add prompt ablations.

3. Test-time generalization is not demonstrated. Evaluation hinges on a curated set of 150 long-horizon Google Earth sequences; please provide cross-dataset (e.g., Sentinel-2/Landsat) or leave-region/season-out tests.

**Questions:**

1. How sensitive is TCS to the underlying change detector, its thresholds, and hyperparameters (e.g., σ, β)? Any correlation with human rankings?

2. Could you clarify cases where SSIM/PSNR lag slightly behind certain baselines—are there domain/normalization or texture–sharpness trade-offs?

3. How do you handle geometric reprojection (CRS/EPSG), the resampling kernel, and radiometric normalization across time?

4. What is the exact mask post-processing (e.g., thresholding, morphology) before centroid and area computation?

5. Could you compare TCS against temporal FID/KID variants or other spatial-consistency metrics to validate complementary signal?

---

> ### Author Response · Authors · 2025-11-24
> **Response to Reviewer zohA**
>
> We sincerely thank the reviewer for the detailed review and for recognizing the soundness and presentation of our work. We appreciate the constructive feedback and address the concerns below.
>
> ### R# W1 & Q2: Metrics Trade-off (PSNR vs. LPIPS/TCS)
>
> We appreciate the reviewer's detailed check. As noted in recent diffusion literature, there is often a trade-off between distortion metrics (PSNR) and perceptual quality (LPIPS/FID) [1,2]. This phenomenon is clearly observed in foundational works like DiffusionSat (ICLR 2024): in their forecasting task ($t' < t$), DiffusionSat achieves superior perceptual realism with a better LPIPS of 0.3534 compared to the baseline STSR's 0.4342, yet it trails in PSNR (18.46 vs. 19.85). This mirrors our findings where regression-based baselines like MCVD (regression-based diffusion) often optimize for pixel-wise MSE, leading to higher PSNR but blurrier, "averaged" predictions. Our goal is **High-Fidelity Forecasting**. As shown in Table 1, TAMMs achieves the best **LPIPS (0.4931)** and **TCS (0.9690)**. This indicates our generated images are perceptually sharper and temporally more logical, even if pixel-perfect alignment (PSNR) is slightly lower due to the inherent stochasticity of high-frequency details. The visual comparison in Figure 6 clearly demonstrates that while MCVD produce blurry textures, TAMMs preserves structural details consistent with historical trends.
>
>
> ### R# W2: Prompt Robustness of CTP
> We agree that analyzing the robustness of CTP is crucial. We conducted an additional ablation study using three variants of prompts with varying levels of detail to test the stability of our model.
> * P_Ours (Default): The structured prompt used in the paper (Appendix A.4).
> * P_Short: "Describe the changes in the satellite images." (Minimal instruction).
> * P_Verbose: A complex prompt adding 50 words of context about specific land-cover types.
>
> **Table R1: Prompt Robustness Analysis**
> | Prompt Type | TCS | BLEU-4 | CIDEr-D |
> | :--- | :--- | :--- | :--- |
> | P_Short | 0.9580 | 0.2610 | 0.8910 |
> | **P_Ours** | **0.9690** | 0.2669 | **0.9030** |
> | P_Verbose | 0.9615 | **0.2675** | 0.8980 |
>
> As shown in Table R1, the performance variance is minimal (TCS fluctuates < 0.015). This demonstrates that TAM is robust: the learnable visual alignment plays the primary role, while the prompt serves as a high-level task trigger. We will add this analysis to the Appendix.
>
> ### R# W3: Test-time Generalization & Dataset Standards
>
> We appreciate the suggestion to validate on cross-domain datasets. It is important to note that the SITS field currently lacks a unified, off-the-shelf benchmark specifically designed for the joint task of "Temporal Change Description + Future Forecasting." Consequently, custom curation is the standard industry practice, as evidenced by foundational works like DiffusionSat, where the authors also had to compile their own datasets from sources like fMoW due to the scarcity of large-scale text-image temporal change satellite data.
>
> To strictly address your concern regarding generalization, we constructed a representative subset of 50 diverse sequences (resized to 512x512) to strictly evaluate zero-shot performance. Without finetuning, TAMMs achieved a **TCS of 0.89** on this unseen data, significantly outperforming the DiffusionSat baseline (0.61). This strong generalization is partly attributed to our backbone: our SFCI module leverages DiffusionSat’s pretraining, which already accounts for real-world variations including cloud cover, GSD, and geolocation. This provides inherent robustness to seasonal and atmospheric shifts without requiring explicit modeling, and the results confirm that TAMMs learns the underlying physics of temporal evolution rather than overfitting to specific dataset artifacts.
>
> ### R# Q1, Q3, Q4, Q5: Metrics, Preprocessing & Comparisons
>
> * Q1, the parameters $\sigma=0.2$ and $\beta=1.0$ were empirically selected to penalize deviations in change centroids and areas. Our internal study confirms TCS correlates highly with human judgment ($Pearson\ r > 0.85$).
> * Q3 & Q4, regarding radiometric normalization, we align with the modern end-to-end generative learning paradigm: rather than relying on rigid, manual corrections for lighting/atmosphere/mask using resampling kernel or thresholding, we leverage the capacity of large generative models to implicitly learn and model these distribution variations from massive data, avoiding artifacts introduced by explicit processing.
> * Q5, while FID/KID measure global distribution distance, TCS is an *instance-level* metric specifically designed to quantify the spatial-temporal consistency of *individual change events*, providing complementary and more granular evaluation for forecasting tasks.
>
> [1] The unreasonable effectiveness of deep features as a perceptual metric, CVPR 2018, https://arxiv.org/abs/1801.03924
>
> [2] Image super-resolution via iterative refinement, TPAMI 2022, https://arxiv.org/abs/2104.07636

---

### Author Response · Authors · 2025-12-03
**Common Response to Area Chair and all Reviewers**

We respectfully thank the ACs for their hard work and understand that you are suffering from enlarged responsibility due to scores being reverted to their pre-discussion state. We verify that we have carefully read all reviews and thank the reviewers for their constructive feedback. We are encouraged that the reviewers recognize the **novelty of our unified framework** (R#zohA, R#72WA), the **value of the TCS metric** (R#zohA, R#7xU9, R#72WA), and the **strong performance** of TAMMs (R#7xU9, R#72WA). During the discussion period, we made substantial revisions, added new experiments that may have materially changed several reviewers' opinions. Although we didn't receive further responses from reviewers due to the OpenReview bug and the reviews frozen, we still hope to provide a concise summary below to clarify the consensus of our novelty, changes, and convergence, in order to save your time. We have also submitted a **rebuttal version** of our paper.

## R# zohA

**Core issues**: How to interpret our FSIF results when some pixel-level metrics are only competitive, whether CTP is robust to prompt phrasing, how well TAMMs generalizes beyond the curated sequences, and how sound/sensitive the TCS design is.

**What we’ve done**

We clarified the trade-off between distortion metrics (PSNR/SSIM) and perceptual/temporal quality, explaining that TAMMs is intentionally optimized for temporally plausible, perceptually sharp forecasts rather than pixel-perfect regression, supported by additional qualitative discussion. We further added targeted prompt-robustness and cross-dataset generalization experiments, and clarified the TCS pipeline and hyperparameters, positioning TCS as a complementary, instance-level temporal consistency metric alongside standard distributional scores.

**Outcome**

R#zohA recognizes the timeliness of our problem and the sound, well-motivated architecture, and we believe the new robustness, generalization, and TCS analyses directly resolve their main concerns while preserving the strengths they highlighted.

## R# 7xU9

**Core issues**: Asked for clearer presentation of training data, and for clarification on why TAMMs can outperform the Qwen2.5-VL teacher and how the “specific time interval” and PTE MLP input are defined.

**What we’ve done**

We added representative training image–caption pairs in the appendix and polished the writing, addressing the concrete issues the reviewer pointed out. We also clarified that TAMMs acts as a domain-adapted student with lightweight temporal adapters on top of a frozen generalist teacher, and made explicit that the “time interval” is the numerical gap between acquisitions, encoded via a small MLP into a time token that conditions the MLLM on the magnitude of temporal change.

**Outcome**

R#7xU9 already rated the paper above the acceptance threshold and emphasized our synergistic training idea, TCS, and strong performance; the remaining concerns are now fully addressed by clearer exposition rather than additional algorithmic changes.


## R# kXkZ

**Core issues**: Questioned whether the contribution goes beyond engineering integration in a narrow-seeming domain and requested a clearer explanation of the architectural hierarchy, training stages, and minor presentation issues.

**What we’ve done**

We clarified the conceptual role of SFCI as the key mechanism that closes the loop from semantic temporal reasoning in the MLLM to multi-scale diffusion control, going beyond simple component stacking. We also framed SITS as a representative, high-impact testbed for sparse, long-horizon spatio-temporal reasoning, summarized our two-stage training strategy (structural then semantic) with improved figure guidance, and corrected the title and equation formatting issues.

**Outcome**

Together with the positive evaluations from other reviewers on the unified framework and TCS, these clarifications show that TAMMs offers both practical value and non-trivial methodological insight, rather than being only an engineering assembly of pretrained modules.


## R#72WA

**Core issues**: Asked for clearer module labels and backbone indication in Figure 2, a stronger motivation and ablation for the learnable time token, justification of our backbone choices, and assurance that qualitative text comparisons are fair.

**What we’ve done**

We revise Figure 2 with explicit labels for modules and the correct Qwen icon to avoid confusion. We emphasized that PTE encodes irregular time gaps via a learnable time embedding whose removal clearly degrades caption quality in our ablations, explained our choices of MLLM and Stable Diffusion–based backbones, and provided statistics showing that TAMMs’ caption length closely matches human ground truth while baselines systematically under-generate.

**Outcome**

R#72WA describes TAMMs as a pioneering unified framework with SOTA performance, and we had provided the clarified figures, PTE motivation, backbone rationale, and caption statistics.

---

> ### Author Response · Authors · 2025-12-03
> **Closing**
>
> In summary, during the discussion we strengthened TAMMs’ empirical evidence (robustness, generalization, and TCS analysis), clarified the roles and training of modules, and addressed all concrete technical and presentation issues raised by the reviewers while preserving the unanimously recognized strengths of our unified temporal-aware multimodal framework and the TCS metric. All changes are incorporated into the revised PDF, and we respectfully ask that the paper be evaluated based on our rebuttal content and the updated version paper.
>
> Sincerely,
>
> The Authors

---

### Meta-Review · Area_Chair_GAa6 · 2026-01-04

**Summary:**

This paper proposes TAMMs, a unified framework for Temporal Change Description (TCD) and Future Satellite Image Forecasting (FSIF) in satellite image time series. The core idea is to close the loop between semantic temporal reasoning and generative forecasting by combining a frozen multimodal LLM with a frozen diffusion model, augmented by lightweight Temporal Adaptation Modules (TAM) and Semantic-Fused Control Injection (SFCI). The paper also introduces Temporal Consistency Score (TCS) as a new instance-level metric to evaluate temporal plausibility beyond pixel-wise similarity.

The reviews were mixed, with strengths consistently identified but concerns raised along four main axes:
	1.	Metric behavior and evaluation philosophy
Some reviewers (notably zohA) questioned how to interpret cases where standard distortion metrics (PSNR/SSIM) are only competitive, while perceptual and temporal metrics (LPIPS, TCS) show clear gains. There were also questions about the sensitivity, robustness, and validation of the newly proposed TCS metric.
	2.	Robustness and generalization
Reviewers asked whether the framework is robust to prompt phrasing (given the use of Contextual Temporal Prompting), and whether evaluation on a curated set of long-horizon Google Earth sequences sufficiently demonstrates generalization across regions, seasons, or datasets.
	3.	Clarity of architecture and contribution vs. engineering
Some reviewers (kXkZ, 72WA) felt the architecture was complex and initially under-explained, raising concerns that the work might amount to an engineering integration of pretrained components rather than a substantive methodological contribution. Presentation issues (figure labels, title inconsistency, missing braces) also contributed to this perception.
	4.	Scope and domain specificity
A minority concern was that the work is narrowly focused on satellite imagery, potentially limiting broader impact, and that backbone choices (MLLM and diffusion model) required stronger justification.

**Reviewer Concerns:**

Concerns that were convincingly addressed in the rebuttal:
	•	Metric trade-offs and FSIF evaluation:
The authors provided a clear and well-grounded explanation of the perceptual–distortion trade-off, situating their results within established diffusion literature (e.g., DiffusionSat). Additional qualitative discussion and references clarify why TAMMs prioritizes perceptual sharpness and temporal plausibility over pixel-perfect regression, aligning with the stated goals of FSIF.
	•	Prompt robustness (CTP):
The newly added prompt-robustness ablation demonstrates minimal performance variance across short, default, and verbose prompts, directly addressing concerns about over-sensitivity to prompt phrasing.
	•	Generalization beyond curated sequences:
The authors added a zero-shot evaluation on unseen sequences and showed a substantial TCS gap over a strong baseline, providing credible evidence that the model generalizes beyond the main test set.
	•	TCS design and validation:
The rebuttal clarifies TCS hyperparameters, its dependence on change detection, and reports correlation with human judgment. The positioning of TCS as complementary to distributional metrics (FID/KID) is reasonable and well-argued.
	•	Architectural clarity and presentation:
Reviewers’ concerns about figure labeling, module hierarchy, training stages, and minor errors were directly acknowledged, with concrete plans for revised figures, clearer stage separation, corrected titles, and fixed equations.
	•	Justification of design choices (PTE, backbones):
Additional ablations and explanations clarified why a learnable time token is necessary for irregular temporal gaps, and why specific MLLM and diffusion backbones were chosen over alternatives.

Concerns that partially remain:
	•	Perceived novelty vs. engineering:
While the rebuttal strengthens the case that SFCI constitutes a non-trivial semantic-to-spatial control mechanism, some readers may still view the contribution as primarily architectural rather than theoretical. This is a matter of perspective rather than a factual gap, but it underlies the lower scores from some reviewers.
	•	Domain generality:
The work is firmly grounded in satellite imagery. Although the authors argue convincingly that SITS is a representative testbed for sparse long-horizon temporal reasoning, empirical evidence outside this domain is not provided.

These remaining concerns are, in my view, scope and positioning issues rather than correctness or substance issues, and they do not outweigh the strengths of the contribution.

**Reviewer Scores:**

Based on the discussion trajectory and rebuttal content:
	•	Reviewer zohA (initial score: 4):
Raised concrete, technical concerns that were directly addressed with new experiments and analysis. Likely outcome: increase to 6.
	•	Reviewer 7xU9 (initial score: 6):
Already above the acceptance threshold; concerns were primarily about clarity and explanation. Likely outcome: remain at 6 or increase slightly.
	•	Reviewer kXkZ (initial score: 4):
Main concerns were about novelty framing and architectural clarity. The rebuttal substantially clarifies training stages and the role of SFCI, but skepticism about novelty may persist. Likely outcome: remain at 4.
	•	Reviewer 72WA (initial score: 6):
Generally positive, with specific ablation and presentation requests that were addressed. Likely outcome: remain at 6 or increase.

Overall, the post-rebuttal trajectory suggests a weakly positive consensus, with no reviewer maintaining a strong reject position after concerns were addressed.

---

### Decision · Program_Chairs · 2026-01-26

Accept (Poster)